# Relation of connectome topology to brain volume across 103 mammalian species

**Maria Grazia Puxeddu**[1]*, **Joshua Faskowitz**[1], **Caio Seguin**[1], **Yossi Yovel**[2], **Yaniv Assaf**[2], **Richard Betzel**[1,3,4], **Olaf Sporns**[1,3,4]

1 Department of Psychological and Brain Sciences, Indiana University, Bloomington, Indiana, United States of America, 2 School of Neurobiology, Biochemistry and Biophysics, Tel Aviv University, Tel Aviv, Israel, 3 Program in Neuroscience, Indiana University, Bloomington, Indiana, United States of America, 4 Program in Cognitive Science, Indiana University, Bloomington, Indiana, United States of America

* mpuxeddu@iu.edu

**Data Availability Statement:** Network data and MATLAB code to reproduce the analysis are available at https://doi.org/10.5281/zenodo.10372945.

## Abstract

The brain connectome is an embedded network of anatomically interconnected brain regions, and the study of its topological organization in mammals has become of paramount importance due to its role in scaffolding brain function and behavior. Unlike many other observable networks, brain connections incur material and energetic cost, and their length and density are volumetrically constrained by the skull. Thus, an open question is how differences in brain volume impact connectome topology. We address this issue using the MaMI database, a diverse set of mammalian connectomes reconstructed from 201 animals, covering 103 species and 12 taxonomy orders, whose brain size varies over more than 4 orders of magnitude. Our analyses focus on relationships between volume and modular organization. After having identified modules through a multiresolution approach, we observed how connectivity features relate to the modular structure and how these relations vary across brain volume. We found that as the brain volume increases, modules become more spatially compact and dense, comprising more costly connections. Furthermore, we investigated how spatial embedding shapes network communication, finding that as brain volume increases, nodes' distance progressively impacts communication efficiency. We identified modes of variation in network communication policies, as smaller and bigger brains show higher efficiency in routing- and diffusion-based signaling, respectively. Finally, bridging network modularity and communication, we found that in larger brains, modular structure imposes stronger constraints on network signaling. Altogether, our results show that brain volume is systematically related to mammalian connectome topology and that spatial embedding imposes tighter restrictions on larger brains.

## Introduction

Evolutionary pressure results in continuous modifications to the morphology and function of mammalian bodies and brains. Extant species exhibit huge diversity in brain anatomy and connectivity, with brain size and volume ranging over several orders of magnitude [1–3]. Understanding how variation in brain size is reflected in cognition and behavior remains one

**Funding:** This research was supported in part by Lilly Endowment, Inc., through its support for the Indiana University Pervasive Technology Institute (https://kb.iu.edu/d/anwt). MGP, JF and OS were supported by NIH/NIMH grant R01-MH122957 (https://reporter.nih.gov/search/NvKSVvlCvU2Orz_Nv9R7jg/project-details/10215267). YY and YA were supported by BSFNSF-NIH Computational Neuroscience program, Grant 2018711 (https://new.nsf.gov/funding/opportunities/collaborative-research-computational-neuroscience). The funders had no role in study design, data collection and analysis, decision to publish, or preparation of the manuscript.

**Competing interests:** The authors have declared that no competing interests exist.

**Abbreviations:** CC, co-classification; CMY, communicability; ED, Euclidean distance; GM, grey matter; HARDI, high angular resolution diffusion imaging; IMD, intramodule connection density; MRCC, multiresolution consensus clustering; MRI, magnetic resonance imaging; NPD, Network Portrait Divergence; NSI, negative search information; SPE, shortest path efficiency; WM, white matter.

of the most challenging goals of evolutionary neurobiology [4–6]. To this end, numerous allometric studies have focused on understanding how brain size scales relative to body size [7–9] and how diverse neuronal substructures contribute to the overall brain volume [10–14].

Further insight into scaling principles of the mammalian brain can be gained from the perspective of network neuroscience [15]. By conceptualizing the brain as a network of anatomically interconnected brain areas, referred to as the connectome [16,17], we can study how its topology (i.e., the relations between nodes—neural elements—and edges—anatomical pathways) scales with brain size. Notably, connectome topology displays nonrandom features, such as modular structure [18]. Modules are groups of highly interconnected nodes that promote locally specialized neural processing [19] and make the network resilient to either internal or external perturbations [20,21], for instance, due to neuronal noise, genetic variation, or injury. The modular structure also fosters a balance between segregation and integration in the network, enabling efficient information processing [22,23] and supporting functional specialization of cortical areas and systems. The inter-areal communication dynamic is constrained by this modular architecture and has been studied from different perspectives, including stylized communication models [24–26], making it possible to predict functional and behavioral outcomes [27,28].

Unlike many other complex systems, anatomical brain networks are spatially embedded and energetically expensive. Not only do brain connections incur material and metabolic costs, but also the total length and density of anatomical connections are constrained by cranial volume [29,30]. Theoretical considerations suggest that these constraints on network cost impact the feasibility of connectivity architectures [31–34]. Comparative and empirical studies support these hypotheses and have provided evidence that variations in brain volume across mammalian species shape the underlying topology of anatomical pathways [35–37]. To date, however, comparative studies have been limited to analyses of a small number of species, mostly comprising mouse, rat, macaque, and human. While informative, these efforts come short of capturing the large range of brain volume variation observed across mammalian taxa and species. In addition, previous studies have not explicitly considered how variation in brain volume is related to the 2 key topological features of brain network organization—modular structure and communication efficiency [21].

Here, we address this question by leveraging the MaMI database [38–40], a diverse set of mammalian anatomical brain networks reconstructed from postmortem anatomical and diffusion magnetic resonance imaging (MRI). The full dataset after data processing comprises 103 different species, covering 12 taxonomic orders and super-orders, whose brain size varies over more than 4 orders of magnitude. Previous studies carried out on this dataset have identified conservation principles that allow overall connectivity and wiring cost to be constant across mammals [38]. Using the same dataset, other studies focused on assessing topological distances between species and taxonomy orders. In [40], Laplacian eigenspectral and local topological features (for instance, clustering coefficient) have been used to assess similarity across species. Results indicated a greater similarity among mammalian brains of the same taxonomy order. Finally, in [39], 16 measures of network distance have been used to assess the (dis)similarity across mammals. The study not only confirmed previous results, showing that mammalian connectomes within the same order are more similar, but also demonstrated the existence of a correlation between network and evolutionary distances derived from phylogenetic trees. Altogether, investigations based on network topology have provided compelling insights into the mammalian brain, highlighting both conservation principles and systematic differences across species that can be traced back to the underlying biology.

Here, we extended these efforts by investigating the relationship between brain network topology and brain volume. To this end, we characterized modular structure and

communication efficiency in the MaMI database, and we observed their changes with brain volume. Collectively, our results demonstrate that modular connectome topology is characterized by scaling principles related to brain volume.

## Results

Scaling laws are ubiquitous in biology. In this paper, we investigated the scaling laws of the mammalian brain and connectome. We leveraged diffusion MRI data from the MaMI dataset [38] and mapped the structural connectivity in 201 mammalian brains from 103 distinct species. After describing changes in brain volume across taxonomy orders and single mammals, we characterized the connectome topology of the mammalian brain, observing how brain volume shapes its modular organization and communication efficiency. A thorough description of the network topological measures under exam can be found in the Methods section, while in Glossary and in Fig 1A, we report the basic definitions and a schematic illustration of them, respectively.

### Glossary

Fundamental concepts and terms used in network neuroscience. More technical details about the network topological measures investigated are provided in the Methods section.

**Networks and brain networks**

A network is a mathematical object used to model complex systems, often biological, like the brain. Under the network conceptualization, the fundamental units of a system (cortical areas in the case of the brain) and their mutual interactions are formalized as a set of nodes connected (or not) by edges.

**Network topology**

Relation between nodes and edges. Network topology can be investigated through measures developed in the field of graph theory, like modularity and communication efficiency.

**Network modules**

Empirical networks, such as brain networks, often exhibit a nonrandom distribution of edges that results in distinct groups of highly interconnected nodes, also named modules, communities, or clusters. Network modules can be defined mathematically by optimizing modularity, a function that measures the strength of the division of the network into modules by comparing the within-module connectivity with a null model.

**Network communication**

How nodes are integrated through sequences of edges, i.e., paths. Models of network communication range from routing protocols to diffusive processes. Cardinal measures falling in this range are as follows: (i) shortest path efficiency, which captures the length of optimally efficient routes; (ii) search information, or the amount of information required from a random walker to travel along the shortest path, which estimates the accessibility of shortest routes in the network; and (iii) communicability, or the sum of the lengths of all the possible paths connecting 2 nodes, which quantifies diffusive broadcasting.

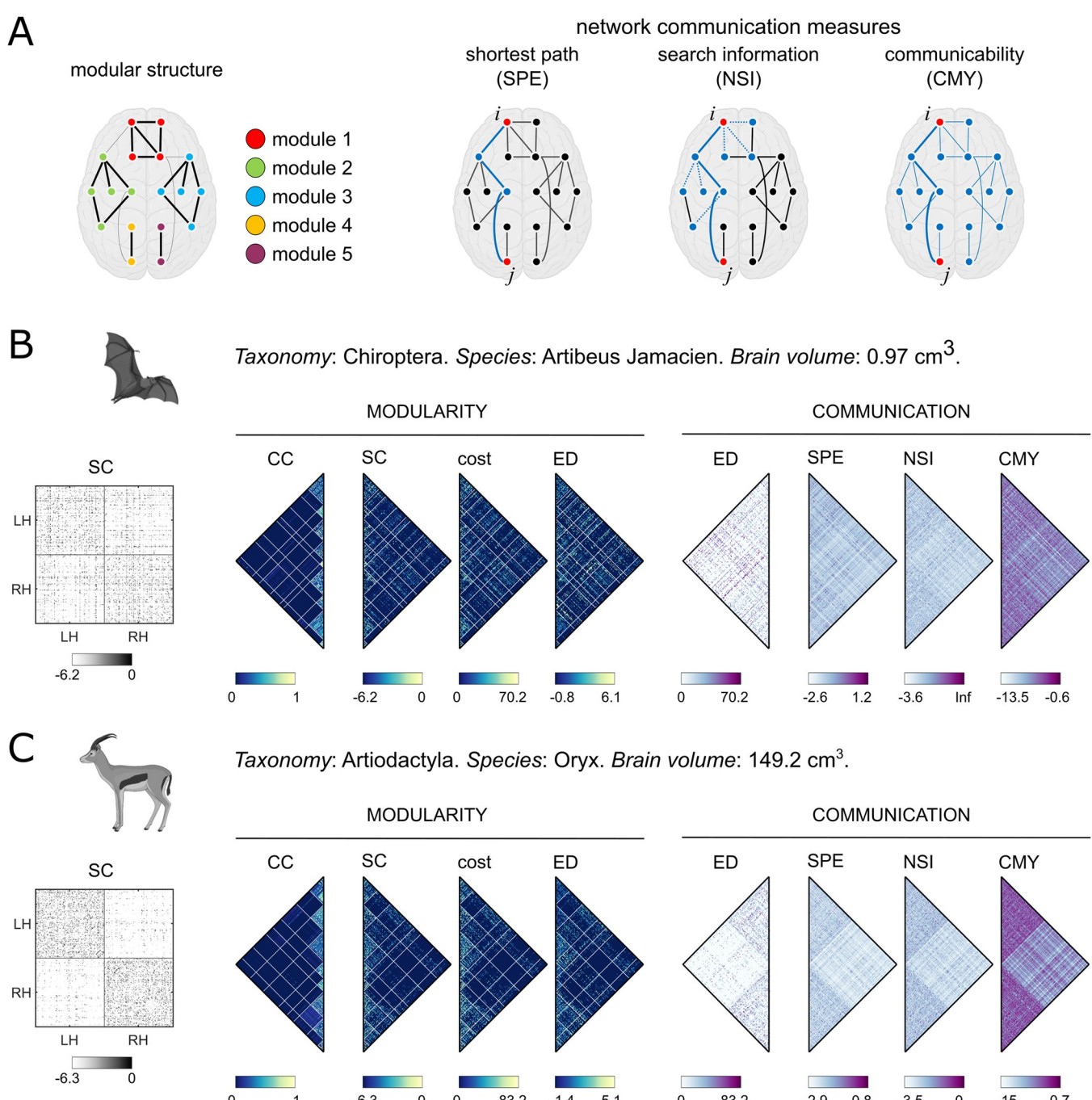

**Fig 1. Schematic representation of the topological analysis.** In panel A, the network topological measures under exams are displayed. On the left, modules of densely interconnected nodes are highlighted through different colors. On the right, 3 putative models to measure network communication efficiency between node *i* and node *j* are illustrated: SPE, or communication via the shortest path; NSI, which also quantifies the dispersion of the signal along the shortest path; CMY, or diffusive broadcast, where all the paths contribute to the network communication efficiency. In panels B and C are reported examples of adjacency and topological matrices for 2 mammals, the bat *Artibeus Jamacien* and the antelope *Orix*. On the left we show the SC matrix. Matrices used for the analysis of the modular structure are the CC matrix (CC), the adjacency matrix (SC), the cost matrix, and the matrix of ED. In this way, the probability of 2 nodes to being assigned to the same module can be compared with the weight and cost of the connection linking the 2 nodes and their physical distance across brain volume. The rows of these matrices have been reordered based on the consensus partition, whose modular borders are displayed as white lines. Matrices used for the analysis of communication efficiency are the ED, the SPE, the NSI, and the CMY matrices. Thus, the physical distance between 2 nodes can be compared to the efficiency with which the signal is transmitted between them according to the 3 different models and across brain volume. Since all matrices are symmetrical, only their upper triangle is shown. The clip arts of the mammals have been generated with BioRender (https://www.biorender.com/). CC, co-classification; CMY, communicability; ED, Euclidean distance; NSI, negative search information; SC, structural connectivity; SPE, shortest path efficiency.

An overview of the data and its topological features is shown in Fig 1B and 1C. Brain volume, together with white matter (WM) and grey matter (GM) volumes, were measured in each mammal to explore the relationship between these 3 anatomical attributes of the brain. Then, focusing on the modular organization of the mammalian connectome, we investigated how the probability of 2 nodes being assigned to the same module (co-classification probability, i.e., CC) relates to the weight (SC) and cost of the anatomical connection linking them (cost), as well as to their distance in the Euclidean space (ED). These relationships were tracked across brain volumes to determine potential scaling laws in modular structure. Other attributes of modular organization were also measured across brain volumes, such as the number of interhemispheric modules and the within-module connection density.

The relationship between network communication efficiency and brain volume was also inspected. We investigated associations between inter-areal Euclidean distance (ED) and communication efficiency, measured in terms of shortest path efficiency (SPE), negative search information (NSI), and communicability (CMY). Again, as we did with the modular organization, the relationships between ED and communication efficiencies were assessed across brain volumes to identify potential scaling laws in network communication.

In the following sections, we present the results obtained from unthresholded brain networks, i.e., unfiltered networks generated by tractography. To show that our results were not biased by possible differences in network density, we report in the Supporting information (S1 File) the results of the same analysis replicated on differently thresholded networks, where we imposed a uniform density $d$ = {0.05, 0.10, 0.15} on each mammalian connectome (density values have been chosen in agreement with previous analyses [39]). Full details about unthresholded and thresholded connectome reconstruction can be found in Methods, Mammalian connectome construction.

## Scaling laws in the mammalian brain

The MaMI dataset comprises images of a broad sample of mammalian brains, the volume of which spans the range [0.0842 $cm^3$, 1597.3 $cm^3$] (Fig 2D). Earlier work [14] has demonstrated that the relation between brain volume, WM volume, and GM volume obeys specific scaling laws. In this section, we studied these relations trying to replicate the same findings on our dataset, in order to validate the imaging approach.

In Fig 2, we report the relations among WM, GM, and brain (BV) volumes across all the mammals present in the dataset. Computing the Spearman correlation, we found all 3 variables to be statistically correlated: $\rho_{GM-WM}$ = 0.9644, $p_{val}$<$10^{-15}$; $\rho_{GM-BV}$ = 0.9979, $p_{val}$<$10^{-15}$; $\rho_{WM-BV}$ = 0.976, $p_{val}$<$10^{-15}$. The regression of WM volume against GM volume in logarithmic space reveals that data points are linearly aligned (Fig 2B), suggesting that their relation approximates a power law. We computed the slope and the intercept of the linear fit using a standard least square approach. In order to limit the weight of outliers, as well as the presence of multiple animals of the same species, we iterated the fitting algorithm in a bootstrap scheme, where each time we considered 90% of the mammals, for a total of 10,000 iterations. This resulted in a mean value and a standard deviation for both the slope and the intercept of the line approximating a scaling law. We obtained the following rule: $\log_{10}$WM = (1.16 ±0.0097)$\log_{10}$GM + (−1.45±0.0445), which means that as the brain size increases, the proportion of WM grows faster than the proportion of GM. This result strongly aligns with the finding established in [14,41], where GM and WM volumes measured on a different mammal dataset have been shown to follow a very similar and equally strong scaling principle. It also

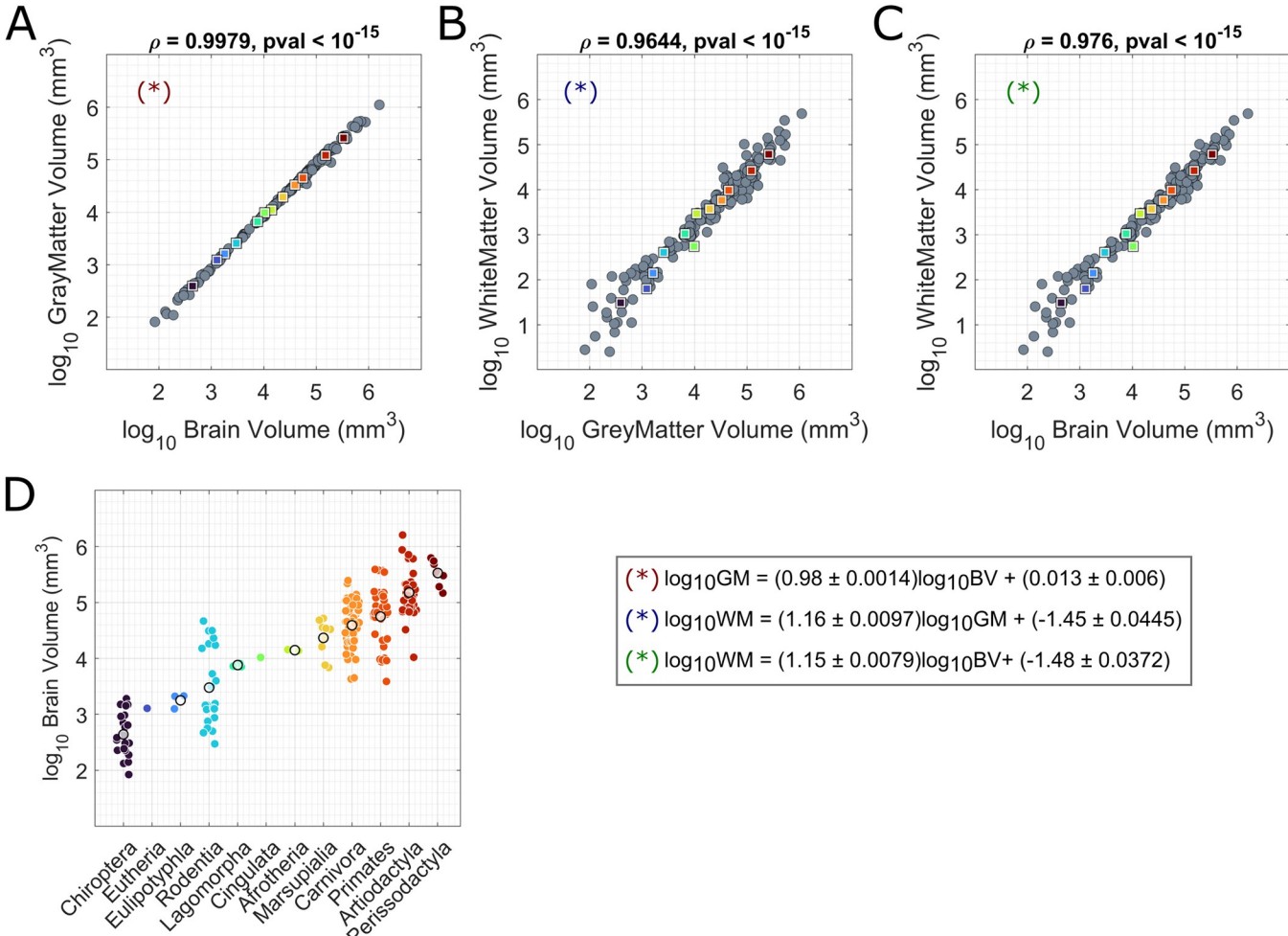

**Fig 2. Scaling laws of the relative sizes of GM and WM volumes in the mammalian brain.** In panels A, B, and C, the relative trends of GM, WM, and total brain volumes are reported; for all 3 volumes, their logarithmic values have been displayed. The colored squares indicate the mean value for each order (colors are matched with panel D). In panel D, the distribution of brain volumes within each taxonomic order is shown. Colored dots refer to single mammals, whereas white dots correspond to the mean value of the distributions. For clades comprising only one mammal, the mean value was not reported. Clades are ordered on the x-axis from lowest to highest average brain volume: $[2.64; 3.11; 3.25; 3.48; 3.88; 4.02; 4.15; 4.37; 4.59; 4.75; 5.18; 5.53]^{10}$mm$^3$ The underlying data for this figure can be found in S1 Data. GM, grey matter; WM, white matter.

confirms the findings in [42], where authors estimated that WM and GM volumes in human and nonhuman primates follow a power law with an exponent close to 4/3.

In addition, GM volume and brain volume also scale approximately as a power law (Fig 2A). Running the same bootstrap procedure as before, we found the following law: $\log_{10}\text{GM} = (0.98 \pm 0.0014)\log_{10}\text{BV} + (0.013 \pm 0.06)$. Once again, this equation aligns with prior studies [14,42].

The scaling laws found in this section confirm previous findings. These results not only validate our mammalian imaging data, but—given the extent of our database—they also corroborate the outcome of classic anatomical and histological observations. Given the existence of these scaling laws, we ask: how do they reflect on the wiring properties of the connectome? Does connectome topology, specifically measures related to network modularity and communication, change systematically with brain size? In the next sections, we tackle these questions, investigating the scaling laws of connectome topology.

## Scaling laws of modular organization

The modular organization of a network generally depicts its tendency to form cohesive, highly interconnected, groups of nodes. Here, we observed how properties of modular organization in the mammalian brain networks vary with brain volume.

To identify the modular structure in each mammalian connectome, we adopted an approach based on multiresolution consensus clustering (MRCC) [43]. Using the Louvain algorithm [44], MRCC optimizes modularity at different spatial scales, delivering multiresolution partitions of the networks into modules of various levels of granularity. More specifically, we imposed that the number of modules ranged between 1 and N/2, with N being the number of nodes. Then, given this set of partitions, MRCC provides a CC matrix whose entries encode the probability that 2 nodes, I and j, belong to the same module, across all spatial scales. MRCC also yields a consensus partition, obtained by clustering the CC matrix. Both the CC matrix and consensus partition offer a way to analyze the modular organization by taking into account a broad spectrum of spatial scales, from very coarse to very fine modules. Here, we present how these properties vary with brain volume. These results are reported in Fig 3. In the Supporting information (S2 File), we reported the same analyses replicated across single spatial scale partitions.

The CC matrix allowed us to analyze the properties of modular organization at a local level, i.e., at the level of nodes and edges. Exploiting the CC matrix (referred to as CC in Fig 1), we investigated how the probability of 2 nodes being assigned to the same community is related to features of the edge connecting them, and how this relationship changes with brain volume. The correlation between CC probability and weight of the connections, $\rho$(CC,weight), was significantly positive in each mammal, meaning that, in the mammalian connectome, the greater the number of streamlines connecting 2 nodes, the more likely it is that those 2 nodes will participate in the same module. To explore how this property varies with the dimension of the brain, we computed the Spearman correlation between the distributions of $\rho$(CC,weight) and brain volume in the dataset. This correlation was statistically significant ($\rho = 0.6$, $p_{val} < 10^{-15}$) (Fig 3A), indicating that the relation between the number of fiber tracts linking 2 nodes and their modular CC is stronger in bigger brains. An analogous relationship emerged comparing brain volume to the correlation between CC probability and cost of the connection $\rho$(CC,cost) (Fig 3C). The cost of a connection takes into account the number of fiber tracts connecting 2 nodes and their length. In every mammalian brain, nodes connected by high-cost edges are more likely to be assigned to the same module. This tendency is much more strongly expressed in bigger brains, as $\rho$(CC,cost) positively correlates with brain volume ($\rho = 0.62$, $p_{val} < 10^{-15}$). The opposite trend is observed when considering the relationship between brain volume and the correlation between CC probability and ED between 2 nodes, $\rho$(CC,ED) (Fig 3B). In every mammal, $\rho$(CC,ED) is significantly negative, meaning that nodes spatially close to each other tend to be assigned to the same module. The correlation between brain volume and $\rho$(CC,ED) is also negative ($\rho = -0.44$, $p_{val} < 10^{-10}$), indicating that being geometrically close is more important for modular CC in bigger brains.

All the correlation coefficients $\rho$(CC,weight), $\rho$(CC,cost), and $\rho$(CC,ED) have been z-transformed before computing the correlation with brain volume. The positive correlation between CC scores and edge weights is expected, because modularity maximization by design delivers modules maximizing the within-module density and strength of connections. However, the novel insight of our analysis is that the magnitude of this correlation varies across brain volume and that the correlation becomes significantly stronger as brain volume increases. These results suggest that spatial embedding of the mammalian brain imposes tighter restrictions on modular organization for larger brains.

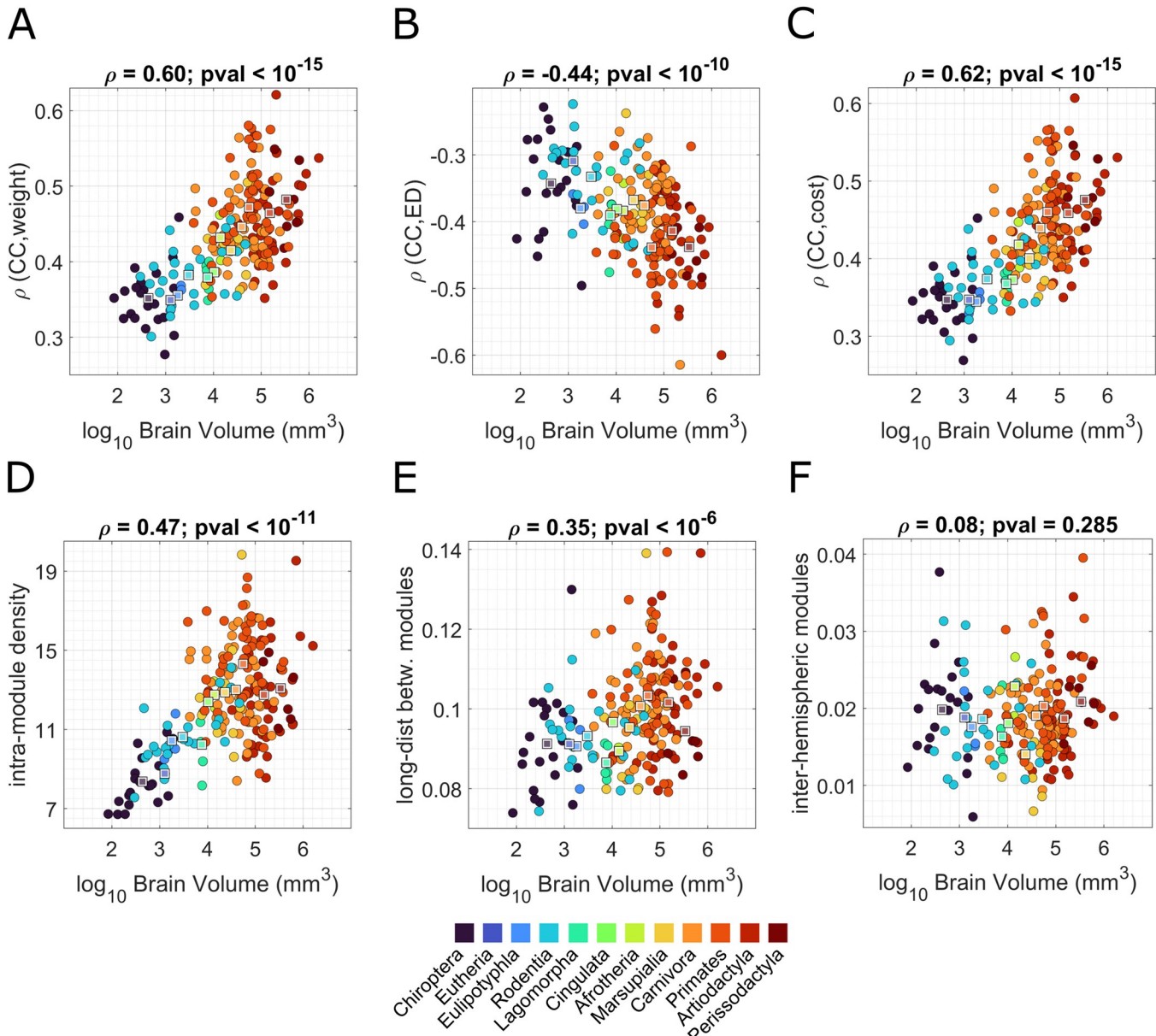

**Fig 3. Modular organization changes with brain volume.** In the 6 panels, we represented the distributions of brain volume (x-axis) versus 6 indices that characterize the modular structure: (**A**) correlation between CC probability and weight of the connection $\rho$(CC,weight); (**B**) correlation between CC probability and ED $\rho$(CC,ED); (**C**) correlation between CC probability and cost of the connection $\rho$(CC,cost); (**D**) within-module density, i.e., average number of connections linking nodes participating in the same modules normalized by module size; (**E**) fraction of long-distance connections used to link nodes belonging to different modules; (**F**) CC probability of nodes belonging to different hemispheres. In each panel, the different taxonomies are highlighted through different colors, and the median values of their respective measures are reported with squares of the same color. The underlying data for this figure can be found in S1 Data. CC, co-classification; ED, Euclidean distance.

We also used the CC matrix to explore the relationship between brain volume and the occurrence of interhemispheric modules. We calculated, for every mammal, the average CC probability of nodes belonging to different hemispheres and examined its trend with respect to brain volume (Fig 3F). We did not find a statistically significant correlation between the 2 variables ($\rho = 0.08$, $p_{val} = 0.285$), meaning that there is no preference in forming interhemispheric modules in smaller or bigger brains.

While the CC matrices allowed us to focus on local topological properties, the consensus partitions obtained for each mammal enabled the characterization of global features of modules that we could link to brain volume. In Fig 3D, we report the correlation between brain volume and intramodule connection density (IMD). The IMD is measured as the average number of connections linking nodes participating in the same modules (normalized by module size), and it is an indicator of the level of segregation in the network. IMD positively correlates with brain volume ($\rho = 0.47$, $p_{val} < 10^{-11}$), suggesting that the modular structure is better defined in bigger brains. A statistically significant and positive correlation ($\rho = 0.35$, $p_{val} < 10^{-6}$) was also found between brain volume and the number of long-distance edges connecting modules (Fig 3E), such that, as brain size increases, the top 5% longest edges (in terms of ED) were progressively more likely to link nodes belonging to different modules.

The results of these correlations were validated by reiterating the analysis on a null model. The same indices reported in Fig 3A–3F were computed on CC matrices and consensus partitions where entries have been randomly permuted 1,000 times (the permutation was consistent across mammals). To build a null model, we used a Mantel test [45]: Rows and columns of the CC matrix were randomized simultaneously while preserving the overall CC distribution. The consensus partitions instead were randomized by permuting module allegiance, while preserving the number and size of modules. All empirical correlations reported here were statistically different from those obtained in the null case (see Supporting information S2 File). Another validation consisted of computing the correlations also on thresholded networks and with respect to GM and WM volumes, instead of total brain volume. Results are consistent with those shown above and have been reported in the Supporting information (S1 File). This does not mean that modularity indices do not change in networks with different densities, but, presumably, they change consistently across mammals, so that their relation with brain volume holds. Finally, given that some species in the MaMI are represented through multiple exemplars, we aimed to show that key results hold even if we consider only one animal per species. We replicated the analyses by randomly selecting one animal per species a thousand times and recomputing the correlations (Supporting information S4 File). Altogether, these control analyses indicate that our results remain robust across a number of methodological factors, suggesting that they are contingent on the empirical topology of the mammalian connectome and not explained by trivial modular structures.

## Scaling laws of network communication

Here, we investigated how communication efficiency varies with the volume of the mammalian brain. Network communication measures quantify how connectome topology supports inter-areal signaling, i.e., the capacity for the integration of functional information across brain regions. While anatomically connected nodes can communicate via direct connections, signaling between unconnected nodes relies on polysynaptic communication mediated by intermediate regions. Multiple models of brain network communication have been proposed, ranging from routing via shortest paths to diffusive broadcasting, but it remains unclear which models most faithfully describe biological neural signaling. To account for this, here, we took into account 3 communication measures, chosen to cover a wide range of possible signaling strategies. Specifically, we considered (i) SPE, modeling communication via selectively accessed shortest paths; (ii) NSI, capturing the ability of diffusion processes to identify shortest paths in the network; and (iii) CMY, a measure of diffusive broadcasting. For each mammalian connectome, each measure yielded a communication matrix of dimension N-by-N, in which entries denote measures of communication between pairs of nodes (Fig 1). For all measures, larger values mean that pairs of nodes can more easily communicate. More details about the

computation of these matrices can be found in Methods, Measures of network communication.

We studied how the physical distance (i.e., ED in Fig 1) between brain regions shapes network communication, and how this relationship between ED and network communication efficiency varies across brain volume. In each mammal, every node pair can be characterized by a physical distance and a network communication estimate (SPE, NSI, and CMY), so that we can compute a correlation between the 2 measures, which we denote as $\rho$(ED,SPE), $\rho$(ED,NSI), $\rho$(ED,CMY). Most of these correlations are negative, suggesting that network communication is more efficient, regardless of policy, between brain regions that are spatially close to each other. Furthermore, the magnitude of them increases with brain volume, i.e., they become progressively more negative, in a statistically significant manner ($\rho = -0.53$, $p_{val} < 10^{-15}$; $\rho = -0.54$, $p_{val} < 10^{-15}$; $\rho = -0.62$, $p_{val} < 10^{-15}$, respectively, for $\rho$(ED,SPE), $\rho$(ED, NSI), $\rho$(ED,CMY)) (Fig 4A–4C). In other words, as the brain size increases, communication becomes progressively more efficient for regions in closer spatial proximity. These results suggest a stronger association between brain geometry and neural signaling in larger brains compared to smaller ones.

Note that in this analysis, we used ED as a measure of distance between nodes, instead of tract length, for 2 reasons. First, anatomical brain networks reconstructed with diffusion imaging and tractography are not fully connected, so fiber length is not available for every pair of nodes. The second reason is more conceptual. While we could have restricted the analysis only to anatomically connected nodes for which we had an estimate of the fiber length, this would have limited the scope of our network communication analyses. Network communication paradigms have been introduced to model and quantify how network topology underpins interareal integration of information among both anatomically connected and nonconnected brain regions, whose signaling relies on polysynaptic paths [24,25].

The correlations presented in Fig 4A–4D were validated by reiterating the analysis on a null model. The same communication measures, and their correlation with the ED between nodes, were computed for each mammal on 1,000 surrogate random networks obtained by preserving the original geometry and degree distribution of the 201 brain networks [46]. Surrogate networks have been generated by binning edges by distance (each bin contained an equal number of edges) and then shuffling edges within bins. Results are reported in the Supporting information (S3 File). All findings reported above for empirical data were statistically different from those obtained in the null case, indicating our results are contingent on the empirical topology and spatial embedding on mammalian connectomes. Furthermore, as we did for the modular structure, we computed the correlations of the 4 indices in differently thresholded networks and with respect to GM and WM volumes, instead of brain volume, finding trends coherent with the ones presented here. Results are reported in the Supporting information (S1 File).

Next, we investigated whether there are preferred strategies of communication in mammalian brain networks, and if those change with brain volume. To this end, we focused on the extremes of the routing-to-diffusion spectrum using SPE and CMY. We brought SPE and CMY to a common, standardized, space by scaling them relative to the average values computed in one-dimensional lattices and random graphs with the same size and density of the brain network [47]. This allowed us to place each mammalian brain in a communication morphospace, whose axes measure standardized network communication via diffusion (CMY$_{scaled}$) and routing (SPE$_{scaled}$) (see Methods). Once placed in a common space, we could observe how SPE$_{scaled}$ and CMY$_{scaled}$ varied across species and brain volume (Fig 4D and 4E). We found that, compared to a null model that combines lattice and random topologies, all the mammalian connectomes are wired so that routing efficiency is always greater than diffusion efficiency (the range of SPE$_{scaled}$ covers higher values). We also found a negative correlation

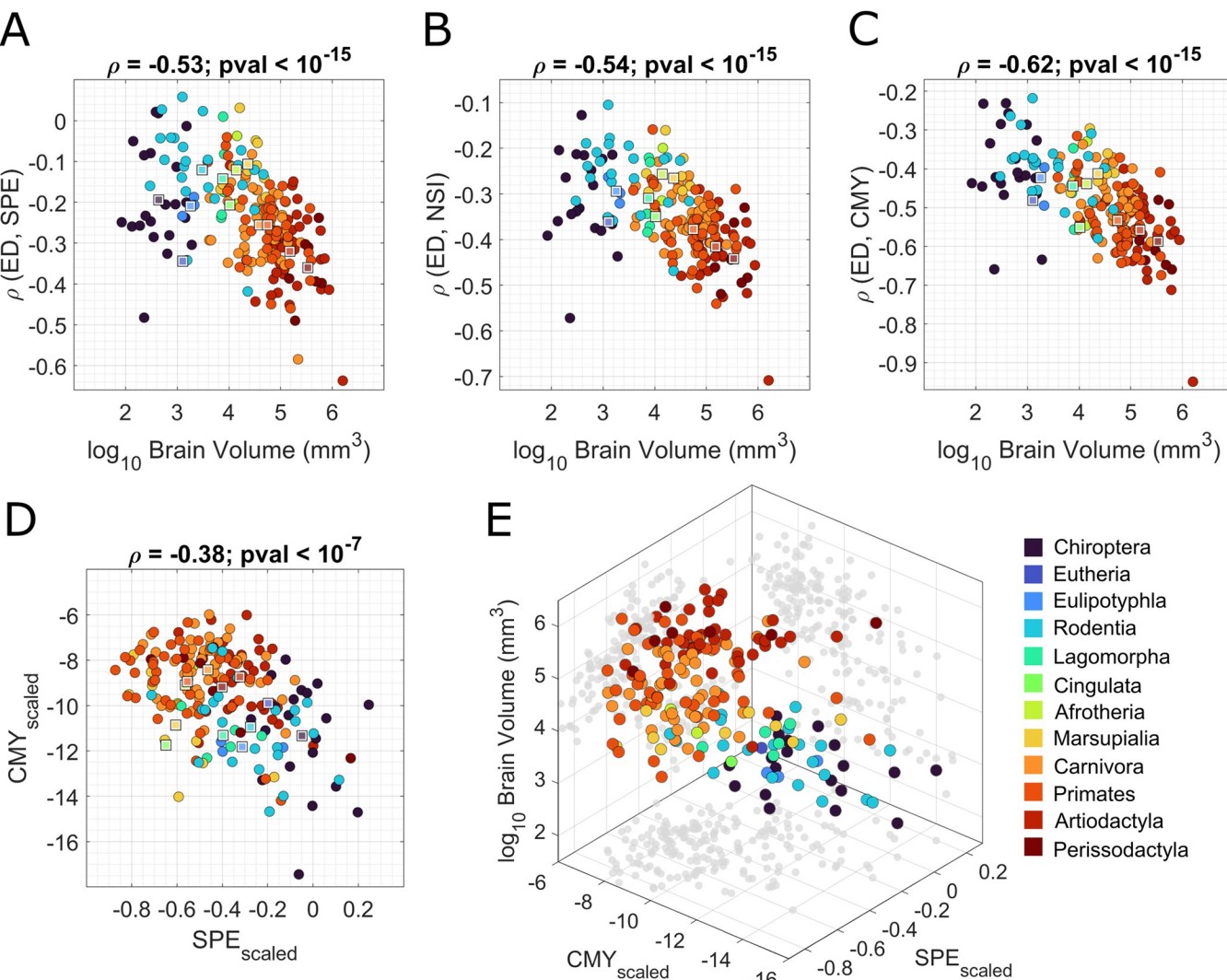

**Fig 4. Network communication efficiency changes with brain volume.** In panels A-C, we represented the distributions of brain volume (x-axis) versus 4 indices that characterize communication efficiency in the brain networks: (**A**) correlation between ED and SPE, $\rho$(ED,SPE); (**B**) correlation between ED and NSI, $\rho$(ED,NSI); (**C**) correlation between ED and CMY, $\rho$(ED,CMY). In panels D and E, we report the mutual trend of SPE$_{scaled}$ and CMY$_{scaled}$ once scaled in a common space, and their trend with respect to brain volume. In each panel, the different taxonomies are highlighted through different colors, and the median values of their respective measures are reported with squares of the same color. The underlying data for this figure can be found in S1 Data. CMY, communicability; ED, Euclidean distance; NSI, negative search information; SPE, shortest path efficiency.

between SPE$_{scaled}$ and CMY$_{scaled}$ ($\rho = -0.38$, $p_{val} < 10^{-7}$)), meaning that species characterized by higher diffusion efficiency display lower routing efficiency and vice versa. More importantly, we observed a gradient in brain communication policies across brain volume. As the brain becomes bigger, diffusion efficiency improves, and, conversely, smaller brains are characterized by higher routing efficiency with respect to bigger brains. These results might suggest that brain volume impacts the connectome topology so that its wiring facilitates signal broadcasting in bigger brains and selective transmission of information in smaller brains.

Lastly, we explored bridges between modular organization and communication dynamics by observing how these properties covary across brain volume. We found that, in all the mammalian species, the CC is positively correlated with all measures of network communication (Fig 5). This indicates that information flow is always more efficient among brain regions that

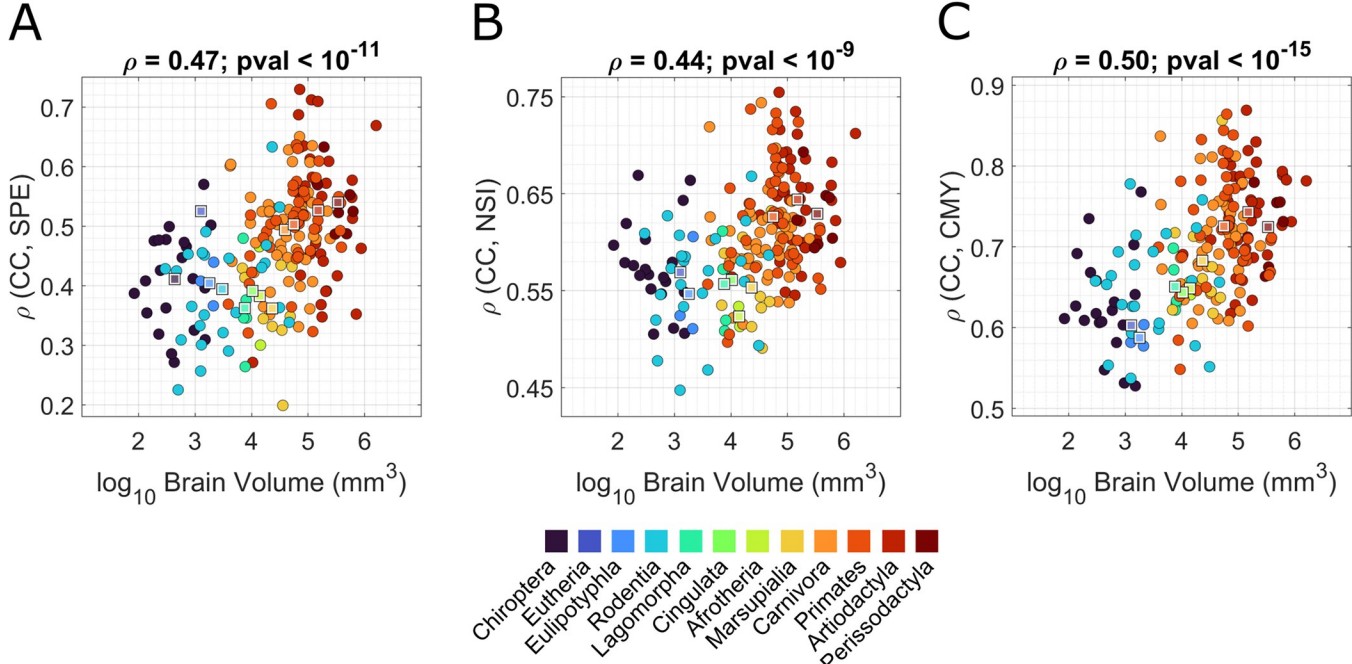

**Fig 5. Bridges between the modular organization and communication efficiency.** In the 3 panels, we represented the distributions of brain volume (x-axis) versus the correlation between CC probability and the 3 indices that characterize the communication efficiency in the brain networks: SPE, NSI, and CMY. In each panel, the different taxonomies are highlighted through different colors, and the median values of their respective measures are reported with squares of the same color. The underlying data for this figure can be found in S1 Data. CC, co-classification; CMY, communicability; NSI, negative search information; SPE, shortest path efficiency.

belong to the same module. More importantly, this correlation scales positively with brain volume ($\rho = 0.47$, $p_{val} < 10^{-11}$, $\rho = 0.44$, $p_{val} < 10^{-9}$, $\rho = 0.5$, $p_{val} < 10^{-15}$, respectively, for $\rho(CC,SPE)$, $\rho(CC,NSI)$, and $\rho(CC,CMY)$). This result implies that the larger the brain, the stronger the constraints imposed by modular connectome topology on neural signaling.

## Discussion

Among mammals, brain size has evolved to exhibit enormous variations in order to accommodate the requirements of different environmental niches [3]. The question is then, what happens to brain network organization when brains get bigger? In this paper, we investigated scaling principles of connectome topology in the mammalian brain. We analyzed modular structure and communication efficiency in an extensive dataset comprising 201 mammals, covering 103 unique species and 12 taxonomical orders, whose brain volume spans 4 orders of magnitude. We found that as brain size increases, the connectome tends to have a better defined and increasingly compact modular organization (Fig 6), and that communication is increasingly efficient between brain regions that are spatially close.

Our work builds on the premise that since all mammalian connectomes are geometrically embedded [48], their connections are subject to constraints due to limited resources such as space, time, and material. Compounding such constraints, neuronal density and axonal projections are associated with significant metabolic cost that is assumed to increase with brain volume, imposing limits on the density and length of the axonal bundles that form edges in the inter-areal connectome [29].

The effect of material and metabolic constraints on brain connectivity as brain size increases can be recapitulated by allometric scaling laws. Allometric scaling generally links one

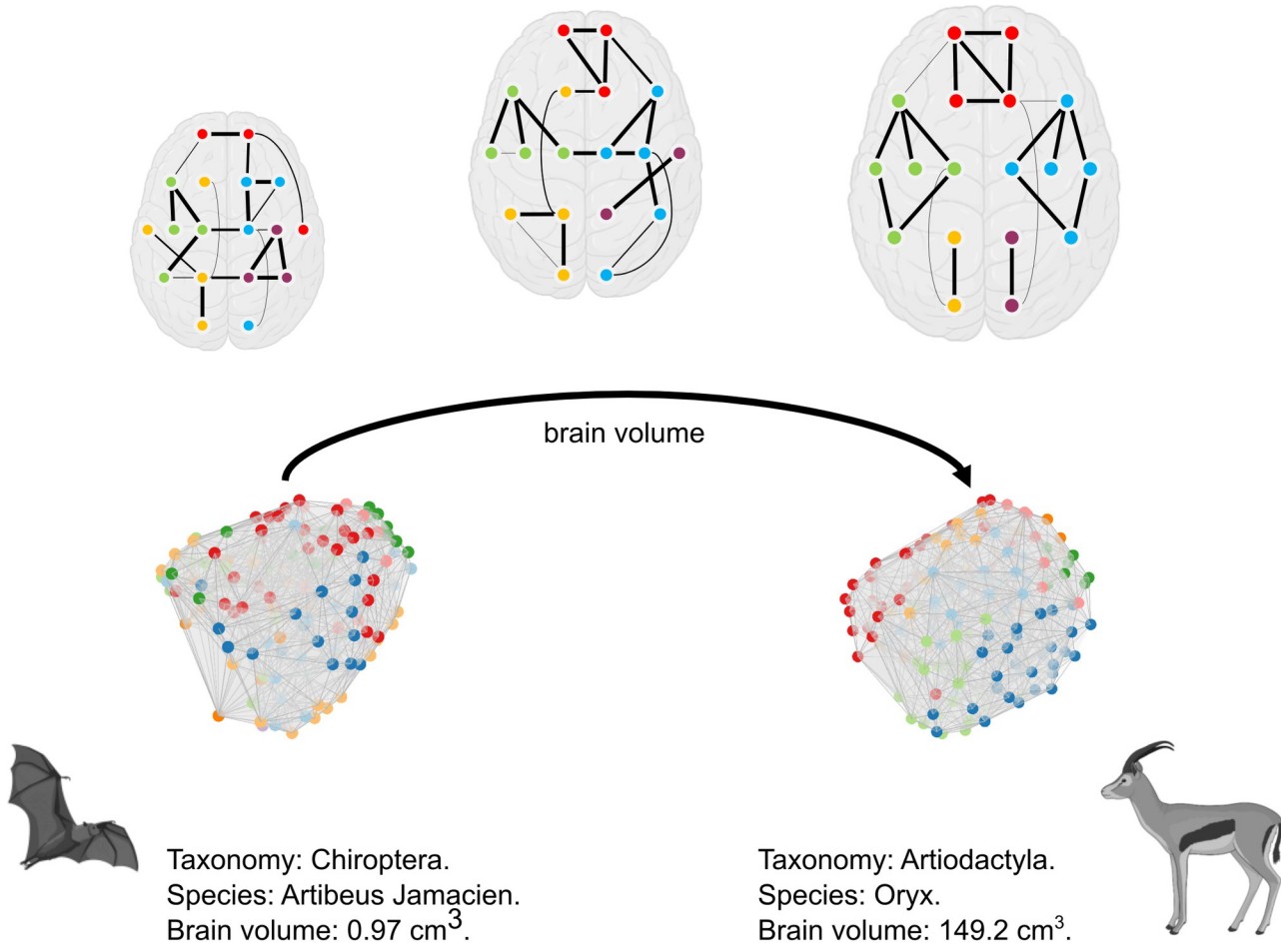

**Fig 6. Schematic representation of how modular organization changes across brain volume.** In the upper half of the figure, we report a toy model showing how as the brain volume increases, modules become more spatially compact and dense, comprising stronger and more costly connections. Long-distance connections are used to connect nodes from different modules rather than nodes coparticipating to the same module. The number of interhemispheric modules does not change with brain volume. In the schematic that we propose, the differences between brain volumes are underscaled for readability purposes, and the gyrification does not mean to reflect any particular species of the dataset. In the lower half of the figure, we display as an example the modular structure of 2 mammalian brains present in the dataset, one small (left) and the other large (right), belonging to the Chiroptera and Artiodactyla orders, respectively. The clip-arts of the mammals have been generated with BioRender (https://www.biorender.com/).

variable to another one through a power law that describes relationships between anatomical or physiological measures that hold true across species of different sizes. They are ubiquitous among biological systems, reflect adaptive and evolutionary processes, and can be found between neural, metabolic, anatomic, cognitive, or behavioral features [49,50].

Through allometric scaling laws, brain size has been used to predict variations in neocortical formations. For instance, while synapse density is invariant relative to brain size, scaling exponents of neuron density, cortical thickness, and WM have been measured to be around $-1/3$, $1/9$, and $4/3$, respectively [51]. In the present study, we computed scaling exponents of GM and WM volumes in the MaMI dataset. We found that GM volume grows proportionally to the overall brain volume, while WM volume grows with an exponent close to $4/3$. Consequently, GM and WM volumes are also linked by a power law with an exponent of $4/3$. These results strongly align with earlier works [14,41,42], so that not only they validate our imaging approach, but also—to date—they represent the most extensive demonstration of these scaling laws in the largest sample of mammalian species.

Other nonconnectomics studies focused on understanding how entire brain components, i.e., anatomical modules, grow in relation to brain size in the mammalian brain. In [11,13], the volumes of the principal substructures (like telencephalon, diencephalon, cerebellum, etc.) have been reported as a proportion of the total brain volume. In [52] instead, the focus was specifically on the primary visual cortex, and scaling laws have been reported describing the proportions of recruited neurons and volume between this neocortical area and the visual thalamus.

In our work, we approach the issue of modular evolution from a connectomics point of view. In fact, not only do neocortical measurements obey allometric scaling laws, but brain connectivity features do as well [36]. In [42], it has been quantified how connectivity asymmetry, clustering coefficient, and characteristic path length change with the overall cerebral volume in 14 primates species. Here, taking advantage of the richness of the MaMI dataset, we pursued a thorough investigation of the relationship between brain volume and connectome topology. Specifically, we were interested in understanding how the modular organization relates to changes in cerebral volume. Modular organization describes the tendency of the network to arrange itself in groups of highly interconnected nodes, allowing for balanced states of segregated and integrated information processing that support brain functioning. It has been observed in the mammalian brain in numerous studies, from macaque [53,54], to cat [23,55], to mouse [56], and to rat [57,58] brains, but so far has not been linked to brain size.

Our results show that brain networks tend to become progressively more modular as the brain volume increases. This result aligns with what has been discussed in [59] from a strictly anatomical point of view. Kaas describes adaptive mechanisms that neocortical structures adopt to cope with the increase of brain volume. These mechanisms include a decrease in proportional connectivity of each neuron with all the others, balanced by an increase of modularity of the cortex, in a way that the distance between neurons that need to interact is reduced. From a network perspective, we found the increase of modularity with brain volume to be reflected in the increment of edge density and weight within modules.

Furthermore, we found that the probability of 2 nodes participating in the same module is statistically associated with the cost and spatial compactness of edges connecting them, and this association scales with brain volume. Specifically, ED between coclassified nodes diminishes with brain volume, whereas the cost of their connection increases. Most of the literature on modular structure of anatomical brain networks reports modules as spatially contiguous. We can hypothesize that spatial compactness reflects adaptive and evolutionary processes that led functionally related brain regions to be positioned close to each other, in order to reduce the cost of wiring, or, alternatively, that prioritized connections among nearby areas that then became more functionally similar. We found that this relation becomes more and more strongly expressed as the brain size increases.

While optimizing anatomical resources leads to a predominance of short-range connections, mostly placed within modules in bigger brains, the connectome is also characterized by a small percentage of long-distance connections that aid global integration and thus support its functionality. Long-distance connections reduce the topological distance in interregional communication [60], render pathways robust to perturbations [61], support functional specificity [62], and link high-degree hub regions forming cores and rich clubs [63–65]. Studies conducted on different species report their role in connecting specialized brain regions with different connectivity profiles [61,62]. Echoing these studies, in our paper, we observed that long-distance connections are used to connect different modules, i.e., specialized groups of nodes. We found that as brain volume increases, more long-distance connections are used to connect different modules.

In [38], the authors used the MaMI dataset to investigate interhemispheric connectivity across mammals, finding that it is overall balanced by intrahemispheric connectivity, so that the global network connectivity is conserved across species. Here, we also observed that the number of interhemispheric connections decreases with brain volume, while the number of intrahemispheric connections increases. Despite the higher interhemispheric connectivity observed in smaller brains (see also S5 File in the Supporting information), our analysis suggests that there is not a significant correlation between brain volume and the probability that 2 nodes of different hemispheres belong to the same module. This might suggest that the way inter- and intrahemispheric connections are organized must compensate for their unequal distribution across brain volume, so that we do not observe statistically significant differences between the probability of having interhemispheric modules in small or big brains.

In this paper, we also investigated scaling laws of communication efficiency. Network communication processes model the transmission of neural signals via structural connectivity [24,25]. There is growing evidence that communication models can explain inter-areal functional connectivity [26,28,66,67] and neural signal propagation [68] and therefore provide a mathematical framework to study communication in brain networks. We modeled interregional information flow in the MaMI dataset using different network measures that account for different strategies of communication, from routing to diffusion. Our results suggest that the physical proximity of brain regions predicts higher communication efficiency and that this prediction is stronger as brain volume increases.

Next, we questioned how different strategies of interregional communication are tied to brain volume. Observing how brain volume, routing efficiency, and diffusion efficiency unravel in a morphospace, we found that smaller brains are characterized by higher routing efficiency with respect to bigger brains, whereas diffusion-based communication becomes increasingly more efficient as the brain volume increases. We speculate that brain volume directly shapes the wiring diagram of the mammalian brain in order to facilitate neural information transmission via signaling broadcasting in bigger brains and via selective pathways in smaller brains. Even if we do not have functional connectivity data to confirm what we speculate, this finding and its interpretation align with [69]. There, the authors analyzed rat, macaque, and human brain data, and, by introducing a parallel communication score, they hypothesize that increasing species complexity and brain volume are associated with a shift of communication policy, from selective (i.e., routing-like) to parallel (i.e., diffusion-like). Moreover, in [68,70], it has been shown that diffusion-based models are well suited to explain functional connectivity and signal propagation after stimulation in nonhuman and human primates. The quest of routing versus diffusion is fairly new in network neuroscience. Our results contribute in demonstrating how the topology of anatomical brain networks changes with brain volume and how it underpins information transmission facilitating signaling diffusion and routing in large and small brains, respectively.

We speculate that, while we built our network communication analyses on undirected networks, the results that we presented are robust in relation to edge directionality, for 2 reasons. First, there are not mathematical reasons for which one communication measure should be hampered more than the others in directed networks. Second, and most important, previous studies demonstrated that even when undirected, the topology of the connectome is pivotal in determining the directionality of information transmission in absence of directed connections [66].

Anatomical modular organization and network communication jointly contribute to a full description of the connectome topology [21], and in this paper, we analyzed how they individually change with brain size. As a last step, we investigated how brain volume shapes their relationship. We found that, across all the species, inter-areal communication efficiency is

positively correlated to modular CC, and this becomes markedly stronger as the brain volume increases.

Collectively, our analyses indicate that the influence of spatial constraints on both network modularity and communication is more pronounced in larger than in smaller brains. Interestingly, diffusive broadcasting is thought to be promoted by a network's modular structure, insofar nodes within a densely connected module can efficiently communicate via diffusion processes [71]. Coupled with observations for the relative preference for diffusion in larger brains, our results lead to the speculation that evolutionary pressure related to spatial constraints, modular architecture, and efficient neural signaling may have coevolved to facilitate broadcasting within modules of specialized information processing [72].

### Technical considerations and limitations

The main limitation of the study concerns of the definition of nodes and edges in the reconstructed brain networks. An ideal comparative analysis would benefit from node correspondence across species, in order to localize and track changes in connectome topology. However, establishing homologies in brain regions of different mammals while accommodating evolutionary specialization is a work in progress [73–75]. Moreover, while most of the efforts in this direction focus on human, macaque, and rat brains, there are many other species in the MaMI dataset that lack canonical or widely accessible anatomical templates. Here, we partially addressed this issue by applying the same parcellation methodology to each mammal. However, while we obtained brain networks with the same number of nodes, brain parcels have been identified through a stochastic procedure based on k-means that impacts node location on cortex and hence edge weights. Thus, we could only examine global properties of modular organization and communication efficiency, without identifying specific neuroanatomical regions that drive changes in connectome topology associated with variations in brain volume. Moreover, because we cannot associate specific brain areas to nodes (i.e., sensory-motor, visual, etc.), we cannot speculate on how our results resonate with established evolutionary theories, like the tethering theory proposed in [76], where changes in brain organization have been associated with the scaling of brain volume in the primate brain. The challenge of node definition, together with other limitations related to WM tracts reconstruction, and hence edge weight interpretation, have been extensively discussed in [39].

Unstable node definition also hampered the possibility of using multilayer models for community detection [77,78], which would be a natural choice in a multivariate dataset. These models have proven extremely useful in network neuroscience to identify community structure across time [79–81], connection modality [82], subjects [83], frequency [84,85], etc. Their utility lies in the fact that they deliver a consistent labeling of the modules across layers, so that one can precisely localize changes in modular partitions. However, they require perfect matching of nodes across layers, so that node i in layer 1 corresponds to the same parcel of the brain in layer 2. This approach is not possible given the data in this study. Thus, instead, we recovered community structure separately in each mammalian brain network using MRCC [43].

An important methodological challenge in the analysis of modularity is the spatial resolution of modules. For any given brain network, the number of modules that can be identified, and, consequently, their size, can vary as a function of a spatial resolution parameter in the modularity optimization algorithm. While a number of heuristics have been proposed to select an optimal value for it, there is no consensus, and different levels of granularity of modular organization have been observed and are equally plausible in the brain. Through MRCC, we considered a broad range of spatial resolutions, analyzing hierarchical structures that mirror our understanding of brain modular organization in the brain. Indeed, neural elements

recursively aggregate to balance information segregation within similar processing units at a single scale and integration between groups of units across multiple scales [86].

As shown in Fig 2D, brain volume and orders are collinear: On average, different orders display different brain volumes, although the interorder variability is high above all among Chiroptera, Rodentia, Carnivora, Primates, and Artiodactyla, and brain volume overlaps across these orders. For this reason, we might expect that the trends here commented could be partially explained by taxonomies and phylogeny. To confirm the robustness of our results and prove that connectome topology is shaped by brain volume, we carried out a supplementary analysis where we computed the partial correlations between modularity/communication indices and brain volume controlling first for the similarity across species and then for taxonomy orders. While several packages are available to estimate phylogeny trees and distances (for instance, [87,88]), we build this analysis on our earlier work [39], where phylogenetic distances were estimated for the species contained in the MaMI database. We report these control analyses in the Supporting information (S6 File). Our results show that, while in some cases they are attenuated, the correlations between network metrics and brain volume remain significant even after phylogenetic distances are accounted for. This indicates that phylogeny and taxonomy only partially explain the observed relationships. Again, this outcome is not surprising given the collinearity observed in Fig 2D between brain volume and taxonomy.

Finally, we observe that while the trends of connectome topology versus brain volume are statistically significant, there is still some residual variance among mammals with roughly equal brain size. This variance could be explained by differences between taxonomies, species, behavioral diversity, or ecological adaptations [89]. The different ways in which the cortex folds across species and orders might also explain why animals with the same brain volume present slightly different features of modular structure and network communication. According to the tension-based hypothesis [90,91], indeed, gyrification is intrinsically intertwined with brain connectivity as the viscoelastic properties of the fiber tracts explain the folding of the brain in gyri and sulci. Furthermore, it has been demonstrated that the degree of cortical folding generally increases with brain size across mammals, but at a different scale between orders and families, and as a function of the cortical thickness in a way that thinner cortices appear more folded [92–94]. Future studies where gyrification or cortical thickness data is available will be needed to uncover how much these factors impact variation of brain network organization across brain volume.

## Conclusion

In conclusion, by analyzing a large collection of mammalian brain networks, covering more than 100 different species, we provided a new insight into the relationship between brain volume and connectome topology. Not only our results corroborate evidences from previous anatomical and histological studies, but also they deliver a new understanding of how the wiring of the mammalian brain evolved to accommodate different sizes.

## Materials and methods

### Ethics statement

All the data used in this work was previously reported in [38], and it is publicly available at https://doi.org/10.5281/zenodo.10372945.

### Mammalian dataset

Our analysis has been carried out on the mammalian MRI dataset (MaMI) [38,40]. It includes ex vivo diffusion and T1- and T2-weighted brain scans from 225 mammals belonging to 125

different species and 12 taxonomic orders (clades): Chiroptera, Eutheria, Eulipotyphla, Rodentia, Lagomorpha, Cingulata, Afrotheria, Marsupialia, Carnivora, Primates, Artiodactyla, and Perissodactyla. No animals were deliberately euthanized for the present study. All brains were collected after incidental deaths of animals in zoos in Israel or natural death collected in the wild. Permission for the collection was granted by the national park authority (approval no. 2012/38645) or its equivalent in the relevant countries. Within 24 hours from death, brains were excised from the skull and fixated through formaldehyde. Fixation required a variable period of time, from a few days to a few weeks, depending on the brain size. Approximately 24 hours before undergoing MRI scanning, brains were immersed in a phosphate-buffered saline solution for rehydration. Limitations of scanning bore made necessary the use of different imaging equipment. Small brains (approximately <0.15 ml) were scanned with a 7 Tesla 30/70 Biospec Avance Bruker system, whereas large brains (approximately >1,000 ml) with a 3 Tesla Siemens Prisma system. Artifacts due to magnetic susceptibility were minimized by placing the brains in a fluorinated oil (Flourinet, 3M) during the scanning session.

Diffusion MRI data were acquired using high angular resolution diffusion imaging (HARDI), which provides a series of diffusion-weighted, spin-echo, echo-planar-imaging images covering the whole brain field of view. The setup for the 7T scanning acquisition included the following imaging parameters: 60 gradient directions, 3 B0 volumes, B-value = 1,000 s/mm$^2$, TR >12,000 ms (depending on the number of slices), TE = 20 ms, $\Delta/\delta$ = 10/4.5 ms. Imaging parameters for the 3T scanning acquisitions included the following: 64 gradient directions, 3 B0 volumes, B-value = 1,000 s/mm$^2$, TR = 3,500 ms, TE = 47 ms, $\Delta/\delta$ = 17/23 ms. All scans were performed using the same 2D pixel grid (128×96), and pixel resolution was linearly scaled with brain size. To keep the signal-to-noise ratio above 20 for all the brains, other scanning parameters were adjusted mammal by mammal. For instance, the number of image slices varied between 46 and 68 to accommodate differences in brain size and shape, and acquisition time approximated 48 hours or 25 minutes in small and large brains, respectively.

For more details about image acquisition protocols and validations on scan parameters, a full report can be found in [38].

## Mammalian connectome construction

The connectome of each brain has been reconstructed from the brain images using the procedure described in [38,39].

WM modeling and tractography were carried out on the MRI data using the software ExploreDTI [95]. Images were preprocessed to reduce the noise (a 3-Pixel Gaussian kernel was applied) and correct for motion, susceptibility, and eddy current distortions. A spherical deconvolutional approach was used to estimate the fiber orientation direction [96], resulting in multiple fiber orientations per voxel. Spherical harmonics of up to fourth order were applied. Whole-brain tractography then was performed using a seed point threshold (fractional anisotropy > 0.2) and half-pixel step length. With this method, approximately 90% of the endpoints resided in the cortical and subcortical GM, avoiding a known bias under which the majority of endpoints inhabit the WM. Finally, streamlines traversing the cerebral peduncle and cerebellar connections were removed. These operations resulted in a list of streamlines for each specimen, with streamline start and endpoints noted.

Downstream of tractography, brain networks were constructed by first parcellating the brain into nodes and then connecting those nodes with weights proportional to the number of streamlines linking them [97]. All brains were subjected to a parcellation procedure that rendered $N$ = 200 nodes. This procedure included manual identification of the brain's midline

and, for each hemisphere, $k$-means clustering of streamlines endpoint positions with $k = 100$. As a result, brains were parcellated into 200 nonoverlapping regions, each one exhibiting a characteristic spatial termination pattern of the streamlines endpoints. Node coordinates were taken to be the center of mass of each node's volume. Fiber length was computed as the average length of all the streamlines connecting nodes. The ED was taken to be the straight-line distance between node coordinates and was measured in voxel units.

The output of the abovedescribed parcellation approach is dependent on the random initial conditions of the $k$-mean clustering algorithm. Thus, for each mammal, 100 runs of the parcellation procedure were carried out, yielding 100 plausible and slightly different instantiations of brain networks for each animal.

The topological analysis proposed in this paper was carried out considering only one representative network for each mammal, estimated from these 100 initial instantiations. To find such a representative instantiation, Network Portrait Divergence (NPD) was employed [98,99]. NPD is a measure used to estimate the distance between 2 networks and is formulated as the Jensen–Shannon divergence between network portraits, with a network portrait being a representation of a network's geodesic, or shortest path, distance. For binary networks, a network portrait is built by deriving the histograms of the matrices thresholded at each shortest path length $l \in 0, \ldots, L$, where $L$ is the diameter of the network, and stacking them in a rectangular matrix $B$, whose entries $B_{(l,k)}$ denote the number of nodes $n$ who have $k$ nodes at distance $l$. For weighted networks, the procedure is the same, but continuous distances need to be discretized into bins. In this study, network portraits were constructed using 25 quantiles to discretize the continuous distances. Bin size was held constant so that distances between networks were comparable across mammals.

For each mammal, NPD was computed between all pairs of the 100 instantiations and stored in a 100-by-100 distance matrix. By averaging the rows of this matrix, average distances from each instantiation to all other instantiations were found. After having removed outliers, i.e., instantiations with an average distance greater than 3 scaled median absolute deviations from the median, the instantiation with the minimal average distance was taken as representative, or centroid, network. Analogously, NPD was also used to identify outliers in the dataset and remove them. The distances between mammalian centroids were stored in a 225-by-225 matrix, and mammals with an average distance from all other mammals greater than 3 scaled median absolute deviations from the median were marked as outliers. Mammals whose network deviated too much from networks of mammals belonging to the same clade were also marked as outliers. To identify them, the 12 taxonomic orders were used to define modules in the 225-by-225 matrix and nodes (i.e., mammals) presenting within-module degree $z$-score greater than 3 were excluded. This process led to a retention of data from 201 mammals.

The NPD was selected to find representative networks and to identify outliers stands based on multiple advantageous features. First, the measure allows for network comparison that without nodal correspondence, allowing for the comparison of mammalian networks for both the 100 network instantiations, and across mammals. Alternative network comparison methods commonly employed, such as the correlation between network upper-triangles, is not possible with the present mammalian data. Furthermore, this measure is based on shortest path, which has been proven to be weakly related to brain volume [38]. Finally, the analysis proposed in [39] demonstrated that NPD computed between mammals correlates with phylogenetic distances between species.

The brain networks can have different densities, which can influence topological measures [100]. To avoid biases in the results due to heterogeneous network density, 3 alternative versions of the dataset have been constructed, where brain networks have been thresholded to a constant density $d = 0.05, 0.10, 0.15$. For each mammal and each one of its 100 network

instantiations, a minimum spanning tree was computed, retaining a skeleton of the N-1 largest edge magnitudes forming a connected component. Then, for each one of the abovementioned thresholds d, edges have been added in order of descending weight until a density d is met. The same procedure based on NPD was applied first to compare different network instantiations finding a representative network for each mammal and then to compare the different mammalian connectomes and identify outliers. As a result, for the 3 thresholds, 3 sets of networks were found, comprising $M$ = {201, 201, 190} mammals.

At the end of data processing, we obtained 4 different datasets corresponding to different network density thresholds: $d$ = {*northreshold*, 0.05, 0.10, 0.15}. The main analysis presented in the Results section was carried out in the first dataset (i.e., no threshold applied to network density). We report in the Supporting information (S1 File) a validation of the results on the thresholded networks. A more detailed description of the connectome reconstruction from MRI scans can be found at [38–40].

## Measures of modular organization

**Multiresolution community detection.**   Assortative communities of the mammalian connectome were recovered through modularity optimization. Given a weighted network $W$, modularity optimization returns the partition that maximizes how much the communities are internally dense with respect to a chance level $P$ [101]:

$$Q(\gamma) = \sum_{ij}[W_{ij} - \gamma P_{ij}]\delta(\sigma_i, \sigma_j). \tag{1}$$

where $P_{ij}$ is the expected weight of the connection between $i$ and $j$ under the null model $P$, $\gamma$ is a resolution parameter, and $\delta(\sigma_i, \sigma_j)$ is equal to 1 when $i$ and $j$ belong to the same community and 0 otherwise. The choice of the null model $P$ is crucial in the optimization and depends on the characteristics of $W$ [102]. For sparse matrices with only positive entries, like the mammalian connectomes, a reasonable and accepted choice is the configuration model, where $P_{ij}$ = $(k_i k_j/2m)$ and is given by the ratio between the product of the degrees of $i$ and $j$ ($k_i$ and $k_j$) and the total number of connections in the network ($2m$). The choice of $\gamma$ instead determines the spatial resolution of the partitions, so that low (high) $\gamma$-values produce coarse (fine) modules.

For each species in the dataset, we recovered partitions at different spatial scales through a 2-step multiresolution approach [43]. In the first step, we logarithmically sampled 500 values of $\gamma$ in the range [0.01 10]. Running the modularity optimization with every one of these $\gamma$-values, we obtained 500 partitions of each connectome. Then, we identified 2 $\gamma$-values, $\gamma_L$ and $\gamma_H$, which produced partitions with at least 2 and maximum N/2 modules (to avoid singleton partitions) for every species. In the second step, we logarithmically sampled 500 values of $\gamma$, again, but this time in the new range [$\gamma_L, \gamma_H$], and we ran the modularity optimization with the new values. In both steps, the modularity optimization has been computed using the function *community_louvain* in the BCT toolbox [103].

Downstream of the 2-step procedure, for every mammal, we obtained 497 differently resolved partitions, from which we computed the CC matrix and the consensus partition. The CC matrix, also called agreement matrix, is a matrix of dimension [N×N], where entries represent the probability that 2 nodes $i$ and $j$ are assigned to the same module, and it is computed as the ratio between the number of the times this condition is verified across all the $\gamma$-spectrum and the total number of $\gamma$-values. The consensus partition, instead, is the partition obtained by running the modularity optimization on the CC matrix [104].

**Features of the multiresolution modular structure.**   We used the CC matrices, or consensus communities, as a basis with which we computed the following measures for every

mammal. Then, we observed their trend with respect to brain volume, WM volume, and GM volume.

**Module density.**   Given a network and its consensus partition into modules, we defined the IMD as the average number of connections within modules, normalized with the dimension of modules.

**Module strength.**   We assessed the strength of the modules by computing the Spearman correlation between the entries of the CC matrix and the adjacency matrix, containing the weight (i.e., strength) of the connections.

**Module cost.**   The cost of the connection linking node i to node j in the connectome is defined as the product between the edge weight and the length of the tract connecting the 2 nodes [29]. We measured the modules cost as the Spearman correlation between the vectors containing the CC probability and connection cost between each pair of nodes.

**Module compactness.**   We assessed the compactness of the modules by computing the Spearman correlation between the entries of the CC matrix and the ED computed between each pair of nodes.

**Long-distance connections between modules.**   Given a network and its consensus modular structure, we computed the tendency of long-distance weights to connect modules as the proportion of the 5% longest edges found between modules, normalized with the total number of modules.

**Interhemispheric modules.**   Given an agreement matrix, we measured the tendency of forming interhemispheric modules as the average CC values between hemispheres normalized with the average CC values within the 2 hemispheres.

## Measures of network communication

**Network communication models.**   Here, we describe the network communication models used in the paper.

**Shortest path efficiency (SPE).**   Given a weighted anatomical network $W$, we can transform edge weights into lengths and obtain a matrix $L$ of structural lengths, by computing the inverse of $W$. $L$ quantifies the travel cost between each pair of nodes [28]. A matrix $\Pi$ of shortest path lengths can be computed by assessing for each pair of nodes $i$ and $j$ the sum of the connection lengths traversed along the shortest path, such that $\pi_{(i \to j)} = L_{iu} + \ldots + L_{vj}$. Then, the SPE is defined as the inverse of this matrix, $SPE = 1/\Pi$ [105].

**Search information.**   The search information measures the bits of information required to traverse a network along its shortest paths [106]. Given $\pi_{(i \to j)}$, the probability that a random walker would take the shortest path traveling from $i$ to $j$ can be computed as $P(\pi_{(i \to j)}) = p_{iu} \times \ldots \times p_{vj}$, where $p_{iu} = W_{iu}/(\Sigma_u W_{iu})$. Search information then is defined as $SI_{ij} = -\log_2[P(\pi_{(i \to j)})]$ [26], with higher values indicating that efficient routes from $i$ to $j$ are less accessible (their cost in bits is high). In this paper, we adopted the NSI, defined as $NSI_{ij} = -SI_{ij} + SI_{ji}$, with the dual purpose of having a symmetrical measure ($SI_{ij} = SI_{ji}$) and an interpretation in line with other measures, where higher values mean higher efficiency.

**Communicability (CMY).**   CMY is defined as the weighted sum of all the possible walks between 2 nodes $i$ and $j$ [107]. For a binary, network is calculated as $G = \sum_{n=0}^{\infty} \frac{W_{ij}^n}{n!} = e^{W_{ij}}$, where $\frac{W_{ij}^1}{1!}$ is the contribution of 1-step walks (i.e., direct links), $\frac{W_{ij}^2}{2!}$ is the contribution of 2-step walks and so on. Weighted networks instead are typically normalized before computing CMY to mitigate the influence of high-strength nodes [108], so that $W\prime_{ij} = W_{ij}/(\sqrt{s_i}\sqrt{s_j})$, with $s_i$ being the strength of node $i$. In this case, $G = e^{W\prime}$.

The abovelisted measures were computed in Matlab through the Brain Connectivity Toolbox [103]. Each one of these models returns a transformation of the brain network into a communication network of dimension [N×N]. To evaluate the influence of the brain volume on these measures, we computed the Spearman correlation between the entries of these matrices and the ED between node pairs. Then, we observed their trends across different brain sizes.

**Morphospace analysis.** We analyzed how brain size affects preferred strategies of network communication, focusing on the communication policies posited at the extreme of the routing-to-diffusion spectrum, SPE and CMY. Because SPE and CMY come with different units, but we wanted to observe their mutual trends, we scaled both measures following the approach proposed in [47]. For each mammal, the average SPE was scaled relative to the average efficiency of the 1D lattices and random networks ($\langle E_{SPE}^{latt} \rangle$ and $\langle E_{SPE}^{rand} \rangle$, respectively) of the same size and density, so that:

$$\|SPE_{scaled}\| = \frac{SPE - \langle E_{SPE}^{latt} \rangle}{\langle E_{SPE}^{rand} \rangle \langle E_{SPE}^{latt} \rangle} \tag{2}$$

Analogously, a scaled version of CMY has been computed and denoted as $CMY_{scaled}$. Once normalized, we could put both $SPE_{scaled}$ and $CMY_{scaled}$ in the same space, termed morphospace, and observe their mutual trends with respect to brain volume.

## Supporting information

**S1 File. Replication on differently thresholded networks.**
(PDF)

**S2 File. Modularity analysis on single-scale partitions.**
(PDF)

**S3 File. Validation of the results on null models.**
(PDF)

**S4 File. Validation of the results on subsamples of the dataset comprising only one animal per species.**
(PDF)

**S5 File. Inter- and intrahemispheric connectivity across brain volume.**
(PDF)

**S6 File. Controlling correlations with similarity between species and orders.**
(PDF)

**S1 Data. Data to reproduce the figures.**
(XLSX)

## Author Contributions

**Conceptualization:** Maria Grazia Puxeddu, Olaf Sporns.

**Data curation:** Joshua Faskowitz, Yossi Yovel, Yaniv Assaf.

**Formal analysis:** Maria Grazia Puxeddu.

**Funding acquisition:** Yossi Yovel, Yaniv Assaf, Olaf Sporns.

**Methodology:** Maria Grazia Puxeddu, Joshua Faskowitz, Caio Seguin, Richard Betzel, Olaf Sporns.

**Supervision:** Olaf Sporns.

**Visualization:** Maria Grazia Puxeddu.

**Writing – original draft:** Maria Grazia Puxeddu.

**Writing – review & editing:** Maria Grazia Puxeddu, Joshua Faskowitz, Caio Seguin, Yossi Yovel, Yaniv Assaf, Richard Betzel, Olaf Sporns.

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
