## [Editor Report · Decision Letter 0]

19 May 2023

Dear Dr Puxeddu, 

Thank you for submitting your manuscript entitled "Network modularity and its relation to brain volume in the mammalian connectome" for consideration as a Research Article by PLOS Biology.

Your manuscript has now been evaluated by the PLOS Biology editorial staff as well as by an academic editor with expertise in neuroimaging and I am writing to let you know that we would like to send your submission out for external peer review.

Once your full submission is complete, your paper will undergo a series of checks in preparation for peer review. After your manuscript has passed the checks it will be sent out for review. To provide the metadata for your submission, please Login to Editorial Manager (https://www.editorialmanager.com/pbiology) within two working days, i.e. by May 21 2023 11:59PM.

Kind regards,

Christian

Christian Schnell, PhD

Senior Editor

PLOS Biology

cschnell@plos.org

---

## [Decision Letter · Decision Letter 1]

7 Aug 2023

Dear Dr Puxeddu,

Thank you again for your patience while your manuscript "Network modularity and its relation to brain volume in the mammalian connectome" was peer-reviewed at PLOS Biology. Please allow me to apologise again for the long delay in sending our decision, which was caused by difficulties in finding reviewers in the first place and then reviewers needing much more time than anticipated. In any case, your manuscript has now been evaluated by the PLOS Biology editors, an Academic Editor with relevant expertise, and by several independent reviewers. 

In light of the reviews, which you will find at the end of this email, we would like to invite you to revise the work to thoroughly address the reviewers' reports.

As you will see below, all reviewers like your study and are largely supportive. Their main concerns are about some of the interpretations and some potentially alternative explanations which can be addressed with textual revisions, a few additional analyses, and improvements of the overall presentation. 

Given the extent of revision needed, we cannot make a decision about publication until we have seen the revised manuscript and your response to the reviewers' comments. Your revised manuscript is likely to be sent for further evaluation by all or a subset of the reviewers.

We expect to receive your revised manuscript within three months. Please email us (plosbiology@plos.org) if you have any questions or concerns, or would like to request an extension. 

**IMPORTANT - SUBMITTING YOUR REVISION**

*Re-submission Checklist*

*Published Peer Review*

*PLOS Data Policy*

*Blot and Gel Data Policy*

Kind regards,

Christian

Christian Schnell, PhD

Senior Editor

PLOS Biology

cschnell@plos.org

REVIEWS:

Reviewer #1: This is a comparative diffusion MRI-based connectomics study that evaluates the relationship between network modularity and brain volumes. The authors analyzed diffusion MRI data acquired from a large number (102) of mammalian species. They first replicated the known scaling law regarding the relationship between gray matter and white matter volume. They then demonstrated how inter-species differences in brain volume are related to the network properties of the connectome. 

I have identified both positive and negative points about the manuscript. Below, I will highlight my comments.

The comprehensive analysis of diffusion MRI data from many mammalian species is an impressive aspect of this study. Additionally, it is nice to see the replication of the scaling law in a neuroimaging dataset acquired from various species. Although this scaling law is widely known, its demonstration in neuroimaging is an important contribution, as it establishes proof that neuroimaging is a useful method for understanding evolutionary principles.

The visualizations presented in this paper are clear, and the analyses are conducted by an experienced group in network neuroscience. The authors have made their MATLAB codes available to the public, which is a good practice for ensuring the reproducibility of findings.

While I don't have significant concerns regarding the scientific goals and reproducibility of this work, I do have some concerns about the writing style and interpretation of findings.

The writing style assumes that readers already possess knowledge about network analysis in connectomics. While this may be suitable for specialized journals in network neuroscience, I believe this style is not appropriate for the current journal, which targets a general readership in biology. The authors introduce many concepts in network analysis, such as modularity and communication efficiency. However, since the introduction does not sufficiently explain the importance of these concepts to general readers, the main results may be difficult to understand for readers who are not experts in network neuroscience. Therefore, if the authors aim to publish this work in this journal, they need to thoroughly revise the paper to make it more accessible to general readers without a strong background in network analysis.

I am concerned that the authors did not acknowledge the works of Karl Zilles and others, who demonstrated a strong relationship between brain volume and gyrification. Without considering gyrification as a confounding factor, interpreting the present results becomes challenging. It is widely reported that tractography can be biased by cortical folding patterns. Hence, I suspect that some inter-species differences in network properties may simply reflect biases in tractography due to sulci/gyri, rather than true biological differences. Conducting control analyses that account for gyrification as a confounding factor is essential to provide confidence in interpreting the results.

Minor point: The citations in this paper do not follow a simple order (1, 2, 3, etc.), making it difficult to track the references within the manuscript.

Reviewer #2: 

This is an interesting manuscript that addresses an important topic in evolutionary neuroscience - how differences in brain volume impact connectome topology. The authors use the MaMI database to identify connectivity modules and examine how connectivity features relate to this modular structure and change with brain size. The authors report that as brain size increases, modules become more compact and exhibit more costly connections, and that distance between notes progressively impacts their communication efficiency. I recommend major revisions to the manuscript, primarily due to the lack of consideration of phylogenetic relationships between species in their statistical analyses - I believe this change can be implemented by the authors. My comments are organized by section below:

Introduction

* Clear and well-written 

Results

* Clear and well-written - great descriptions of approach

* Readers would benefit from added definitions of relevant terms in the first section (e.g., NSI, communicability, etc.), even though these are provided later in the section - definitions could also/alternatively be included in Figure 1

* In some areas, it would be helpful to provide a plain language interpretation of results (e.g., as brain size increases, the proportion of white matter increases faster than the proportion of grey matter, so larger brains are comprised of more white:grey matter; describe grey vs. white matter; etc.) -this is generally well-done throughout 

Figures

* Figure 1 - In its current form, this figure isn't super useful for comprehending what the different measures are capturing - I suggest the authors recreate this figure to add additional information

* Figure 2 - Please color code the dots in A and B as in C; It would be helpful to add a plot showing white matter volume ~ brain volume

Discussion

* Clear and well-written 

* How do these results relate to established hypotheses of brain evo-devo (e.g., the tethering hypothesis: Buckner and Krienen 2013, Trends in Cog Sci)?

Methods

* Species that are more closely related to each other are more likely to be similar to each other - this observation is actually noted in the authors' descriptions of previous analyses of this dataset in the Introduction. Accordingly, phylogenetic comparative methods are necessary to control for the lack of independence (i.e., due to evolutionary relatedness) between data points (i.e., species). These methods can and should be implemented here, including:

-- Phylogenetic correlations (e.g., to replace the reported brain volume vs. grey matter volume correlation)

http://blog.phytools.org/2017/08/pearson-correlation-with-phylogenetic.html

-- Phylogenetic generalized least squares regression modeling (e.g., to replace regressions of grey matter volume ~ brain volume; modular organization measures ~ brain volume; etc.)

The authors can use e.g., the phytools R package and incorporate a mammal-wide phylogenetic tree (e.g., TimeTree), although some effort will likely be required to match species names between the dataset and tree; I would also consider running analyses with order included as a predictor to account for / identify potential inter-order variation in scaling relationships 

I hope the authors find my comments useful in revising their manuscript.

Reviewer #3: ~~~~~~~~~~Summary~~~~~~~~~~

In general, comparative analysis of brain networks across species is greatly limited by the difficulty in alignment of nodes across species, that is, the designation of homologous regions. In their innovative study, the authors have bypassed this issue by focusing on network measures which are agnostic to the identities of individual nodes. Broadly, the claim is made that structural brain networks become more modular with increasing size. This hypothesis is logical giving the increasing volumetric constraints and metabolic burdens of non-modular wiring with size, but to date has not been adequately tested. 

To examine this question, the authors performed diffusion MRI fiber tractography (previously reported) on ~200 ex-vivo examples of ~100 species from 12 orders. Mesoscale nodes were derived by clustering these tracts and the relationships, specifically module cohabitation probability, among nodes quantified by subsequent multi-scale community detection. The amount of variance in module cohabitation of two nodes explained by the volumetric and metabolic cost of the connection between them appears strongly predictive total brain volume. Similarly, brain volume appears strongly predictive of the amount of variance in internodal communicative capacity explained by internodal distance and the amount of variance in internodal module cohabitation explained by internodal communicability.

In whole the results shown support the author's claims. The analyses performed are rigorous, figures mostly clear, and the abstract measures well explained. My first two comments concern principled quantitative adjustments that I do not anticipate will dramatically change the tenor of the results. The third presents an alternative to the volumetric hypothesis which may call for additional analyses and certainly calls for additional discussion. The remaining comments highlight underexamined elements for further discussion.

~~~~~~~~~~Major concerns~~~~~~~~~~

1. It is unusual to calculate correlation coefficients of correlation coefficients. One reason is that the sample distributions of coefficients become progressively skewed as the mean coefficient approaches 1 or -1. This can be resolved by applying Fisher's z-transformation to primary coefficients before correlating them with brain volume to yield secondary coefficients. While this adjustment is principled and should be performed, I do not believe it will change the study's findings, as the reported primary coefficients are clustered reasonably tightly and do not approach the [-1 1] limits closely, and because the secondary correlations were calculated with the robust Spearman rank-order procedure. 

2. It is unclear why the Euclidean distance between nodes was used when inter-node fiber length was estimated and used for the edge cost calculation. Fiber length is surely a more biologically meaningful proxy for the closeness of node and modules, especially as, at least in macaques, fiber length is not anticorrelated to conduction velocity (Innocenti et al., 2014). Furthermore, in gyrencephalic cortices, Euclidean paths often implausibly transect sulci. Summarily, when examining communication measures (Fig. 4), cost rather than Euclidean distance alone seems more appropriate, though given the similarity of Figs. 3 A, B, C the conclusions will almost certainly hold. 

3. There is no discussion of the differing degrees of gyrification among the examined taxa. Given that that gyrification is correlated to brain weight (and presumably volume) across species (Mota & Herculano-Houzel, 2015; Pillay & Manger, 2007), is it possible the observed trends are driven by or at least modulated by gyrification. Would the figures be similar if an index of gyrification was substituted for volume? There is considerable variation in the network metrics at a given brain volume; can some of it be explained by gyrification? 

4. The control analyses performed for random networks and fixed network density are welcome. Is there any relationship between un-thresholded network density and brain volume, and how do the unthresholded densities quantitatively compare to the thresholds? Given that long distance connections are much weaker than short range connections, one would expect thresholding to increase modularity. The fact that key findings are not affected by imposing very strict density threshold requires further explanation. 

5. Could the authors speculate on why some the species appear to markedly deviate from the observed trends? For example, there appears to be one marsupial and one chiropteran which stand out in their modularity and communication efficiency. They may have unique anatomical or ecological adaptations which may serve as enriching counterexamples. 

6. The insensitivity of MRI tractography to the directionally of connections is well known. Does the possible asymmetry of edges have implications when discussing routing- vs diffusion- signaling strategies? Prima facie, it seems obscuring true edge asymmetry would bias toward diffusive strategies. 

7. While the relationship between mesoscale structural and functional connectivity is fraught (Honey et al., 2009), what functional predictions can be made from result from authors' findings, e.g., "wiring facilitates signal broadcasting in bigger brains and selective transmission of information in smaller brains". Is there evidence for these predictions in the mouse and human functional literature (being better sampled examples of small and big brains). Similarly, how does anatomical spatial compactness square with wide spatially noncontiguous functional networks, e.g., default mode and language processing (Ji et al., 2019)?

~~~~~~~~~~Minor concerns~~~~~~~~~~

The number of digits used for rho values throughout the manuscript is inconsistent. 

Citation 82 needs to be updated from the preprint to the published article. It's probably a good idea to check the other preprint citations as well. 

Fig. 1: The colormaps are diverging, but the mid-point of the colorscale is not meaningful (e.g., 0)

Fig. 3: Panels B and C should be reversed as weight and length are constituents of cost.

Fig. 4: The direction the SPE axis is reversed in panel D vs panel E. Also, unlike the CMY axis, the ticks are not consistent between the panels. Also, the descriptions of panels B and C are reversed in the legend relative to the to figure and the abbreviation "co." is used instead of "CMY". 

Fig. 6: I know the cortices at the top of the figure are meant to be illustrative, but they overemphasize the effect of brain volume by (1) showing an under-scaled size difference, and (2) presenting the same shape and folding pattens as the same in the different scaled brains. 

There are a few very minor typos / grammatical errors in the manuscript, e.g., not pluralizing 'cost' in the second sentence of the abstract, substituting 'tour' for 'our' in the first paragraph of Pg.6 Col. 2, substituting 'constrains' for 'constraints' twice in the last paragraph of Pg. 11 Col. 1. 

~~~~~~~~~~References~~~~~~~~~~

Honey, C. J., Sporns, O., Cammoun, L., Gigandet, X., Thiran, J. P., Meuli, R., & Hagmann, P. (2009). Predicting human resting-state functional connectivity from structural connectivity. Proceedings of the National Academy of Sciences of the United States of America, 106(6), 2035-2040. https://doi.org/10.1073/pnas.0811168106

Innocenti, G. M., Vercelli, A., & Caminiti, R. (2014). The Diameter of Cortical Axons Depends Both on the Area of Origin and Target. Cerebral Cortex, 24(8), 2178-2188. https://doi.org/10.1093/cercor/bht070

Ji, J. L., Spronk, M., Kulkarni, K., Repovš, G., Anticevic, A., & Cole, M. W. (2019). Mapping the human brain's cortical-subcortical functional network organization. NeuroImage, 185(September 2018), 35-57. https://doi.org/10.1016/j.neuroimage.2018.10.006

Mota, B., & Herculano-Houzel, S. (2015). Cortical folding scales universally with surface area and thickness, not number of neurons. Science, 349(6243), 74-77. https://doi.org/https://doi.org/10.1126/science.aaa9101

Pillay, P., & Manger, P. R. (2007). Order‐specific quantitative patterns of cortical gyrification. European Journal of Neuroscience, 25(9), 2705-2712. https://doi.org/https://doi.org/10.1111/j.1460-9568.2007.05524.x

---

## [Decision Letter · Decision Letter 2]

1 Dec 2023

Dear Dr Puxeddu,

Thank you for your patience while we considered your revised manuscript "Network modularity and its relation to brain volume in the mammalian connectome" for publication as a Research Article at PLOS Biology. This revised version of your manuscript has been evaluated by the PLOS Biology editors, the Academic Editor and the original reviewers.

Based on the reviews and on our Academic Editor's assessment of your revision, we are likely to accept this manuscript for publication, provided you satisfactorily address the following data and other policy-related requests.

* We would like to suggest a different title to improve readability/accuracy: "Brain volume is systematically related to connectome topology across 103 mammalian species"

* Please provide the links to the funding bodies in the Financial disclosure section in the manuscript details.

* Thanks for providing the approval number from the Isreal National Park authority. Please provide the authorities and approval number for the other countries as well. Depending on the number of countries, you could provide these in a table.

DATA POLICY:

Regardless of the method selected, please ensure that you provide the individual numerical values that underlie the summary data displayed in the following figure panels as they are essential for readers to assess your analysis and to reproduce it: Figure 2, Figure 3, Figure 4, Figure 5, and Figure S5

CODE POLICY

Per journal policy, as the code that you have generated is important to support the conclusions of your manuscript, we require that you make it available without restrictions upon publication. Please ensure that the code is sufficiently well documented and reusable, and that your Data Statement in the Editorial Manager submission system accurately describes where your code can be found.

Please also generate a DOI for your data repository. One way to achieve this is using zenodo.

We expect to receive your revised manuscript within two weeks. 

*Published Peer Review History*

*Press*

Sincerely,

Christian

Christian Schnell, PhD

Senior Editor,

cschnell@plos.org

PLOS Biology

Reviewer remarks:

Reviewer #1: I thank authors for addressing my comments. I have no further comments.

Reviewer #2: I am satisfied with the authors' responses.

Reviewer #3: The authors have addressed my concerns and the revised manuscript is suitable publication.

---

## [Editor Report · Decision Letter 3]

8 Jan 2024

Dear Dr Puxeddu,

Happy New Year!

Thank you for the submission of your revised Research Article "Relation of connectome topology to brain volume across 103 mammalian species" for publication in PLOS Biology. On behalf of my colleagues and the Academic Editor, Hiromasa Takemura, I am pleased to say that we can in principle accept your manuscript for publication, provided you address any remaining formatting and reporting issues. These will be detailed in an email you should receive within 2-3 business days from our colleagues in the journal operations team; no action is required from you until then. Please note that we will not be able to formally accept your manuscript and schedule it for publication until you have completed any requested changes.

PRESS

Sincerely, 

Christian

Christian Schnell, PhD

Senior Editor

PLOS Biology

cschnell@plos.org